# Peripheral Inflammatory Indexes Neutrophil/Lymphocyte Ratio (NLR) and Red Cell Distribution Width (RDW) as Prognostic Biomarkers in Advanced Solitary Fibrous Tumour (SFT) Treated with Pazopanib

**DOI:** 10.3390/cancers14174186

**Published:** 2022-08-29

**Authors:** Samuel Hidalgo-Ríos, Jaime Carrillo-García, David S. Moura, Silvia Stacchiotti, Antonio López-Pousa, Andrés Redondo, Antoine Italiano, Antonio Gutiérrez, Giovanni Grignani, Nadia Hindi, José-Antonio López-Guerrero, Xavier García del Muro, Javier Martínez Trufero, Emanuela Palmerini, Ana Sebio García, Daniel Bernabeu, Axel Le Cesne, Paolo Giovanni Casali, Jean-Yves Blay, Josefina Cruz Jurado, Javier Martin-Broto

**Affiliations:** 1Department of Clinical Pharmacology, University Hospital Virgen del Rocío, 41013 Sevilla, Spain; 2Health Research Institute Fundación Jiménez Díaz (IIS-FJD, UAM), 28040 Madrid, Spain; 3Medical Oncology Department, University Hospital General de Villalba, 28400 Madrid, Spain; 4Adult Mesenchymal and Rare Tumor Unit, Department of Cancer Medicine, Fondazione IRCCS Istituto Nazionale dei Tumori, 20133 Milan, Italy; 5Department of Medical Oncology, 08041 Sant Pau Hospital, Barcelona, Spain; 6Department of Medical Oncology, University Hospital La Paz, Institute for Health Research (IdiPAZ), 28046 Madrid, Spain; 7Department of Oncology, Institut Bergonié, 33000 Bordeaux, France; 8Department of Hematology, University Hospital Son Espases, 07210 Palma, Spain; 9Sarcoma Unit, Division of Medical Oncology, Candiolo Cancer Institute, 10060 Candiolo, Italy; 10Medical Oncology Department, University Hospital Fundación Jiménez Díaz, 28040 Madrid, Spain; 11Laboratory of Molecular Biology, Fundación Instituto Valenciano de Oncología, 46009 Valencia, Spain; 12Catalan Institute of Oncology, Institut d’Investigació Biomédica de Bellvitge, University of Barcelona, 08907 Barcelona, Spain; 13Department of Medical Oncology, University Hospital Miguel Servet, 50009 Zaragoza, Spain; 14Chemotherapy Unit, Istituto Ortopedico Rizzoli, 40136 Bologna, Italy; 15Musculoskeletal Imaging Section, University Hospital La Paz, 28046 Madrid, Spain; 16Department of Medical Oncology, Gustave Roussy Cancer Campus, 94805 Villejuif, France; 17Department of Medical Oncology, Centre Léon Bérard, Université Claude Bernard Lyon, 69008 Lyon, France; 18Department of Medical Oncology, University Hospital of Canarias, 38320 Tenerife, Spain

**Keywords:** solitary fibrous tumour, pazopanib, inflammation, neutrophil/lymphocyte ratio, NLR, platelet/lymphocyte, PLR, red cell distribution width, RDW

## Abstract

**Simple Summary:**

Pazopanib treatment in advanced solitary fibrous tumour patients, assessed in the prospective GEIS-32 phase II clinical trial, has shown longer progression-free survival and overall survival versus chemotherapy treatment in control patients. In recent years, the interest in the prognostic and predictive value of different peripheral inflammatory indexes, such as neutrophil/lymphocyte ratio, platelet/lymphocyte ratio, and red cell distribution width, has been increased in sarcomas, showing significant results in different soft tissue sarcomas. However, they have not been previously analysed in solitary fibrous tumour (SFT) patients. These indexes were retrospectively analysed in the typical- and malignant-SFT cohorts treated with pazopanib of the GEIS-32 trial to evaluate their predictive or prognostic value.

**Abstract:**

Pazopanib was assessed prospectively in the GEIS-32 phase II study (NCT02066285) on advanced solitary fibrous tumour (SFT), resulting in a longer progression-free survival (PFS) and overall survival (OS) compared with historical controls treated with chemotherapy. A retrospective analysis of peripheral inflammatory indexes in patients enrolled into GEIS-32 was performed to evaluate their prognostic and predictive value. Patients received pazopanib 800 mg/day as the first antiangiogenic line. The impacts of baseline neutrophil/lymphocyte ratio (NLR), platelet/lymphocyte ratio (PLR), and red cell distribution width (RDW) on PFS, OS, and Choi response were evaluated by univariate and multivariate analysis. Metastasis-free interval (MFI), mitotic count, and ECOG were also included as potential prognostic factors. Sixty-seven SFT patients, enrolled in this study, showed a median age of 63 years and a female/male distribution of 57/43. The median follow-up from treatment initiation was 16.8 months. High baseline NLR, PLR, and standardised RDW were significantly associated with worse PFS and OS. NLR, RDW, MFI, and mitotic count were independent variables for PFS, while RDW and ECOG were independent for OS. Further, NLR and mitotic count were independent factors for Choi response. High baseline NLR and RDW values were independent prognostic biomarkers for worse outcome in advanced SFT patients treated with pazopanib.

## 1. Introduction

Solitary fibrous tumour (SFT) is a rare and ubiquitous fibroblastic mesenchymal neoplasm with a specific histological architecture, which harbours an intrachromosomal NAB2/STAT6 fusion gene and shows nuclear immunoreactivity for STAT6 [1]. Typical and malignant SFT subtypes were differentiated in the WHO 2013 classification of soft tissue tumours, taking into account the number of mitoses per 10 high-power fields, the presence of necrosis, and nuclear pleomorphism. However, despite the differences in these parameters, up to 45% of cases in both subtypes can develop metastasis. Thus, the last WHO 2020 classification of soft tissue and bone tumours advises the use of risk stratification models to determine prognosis in SFT, based on age, tumour size, and number of mitoses or even with the presence of necrosis [2]. Dedifferentiated SFT is a more rare and aggressive subtype, which shows a fast transition to a high-grade sarcoma. The most frequent localisations of SFTs are abdominal and thoracic cavities, followed by limbs and intracranial locations. Surgical resection is the preferred treatment for localised disease (>90% of cases), showing a 10-year overall survival (OS), which ranges from 54% to 89%. However, locally advanced or metastatic disease is treated with conventional chemotherapy, although it has limited efficacy [3,4]. Pazopanib is an antiangiogenic drug that has shown longer progression-free survival (PFS) and OS in advanced SFTs compared with chemotherapy in historical controls [5,6]. Pazopanib was approved in second lines of advanced soft tissue sarcomas (STS) based on a significant improvement in PFS compared with placebo [7]. However, the identification of prognostic biomarkers of pazopanib response seems necessary to improve the clinical effectiveness and cost-effectiveness of this drug.

Pazopanib is a tyrosine kinase inhibitor (TKI) targeting VEGFRs, PDGFR α/β, FGFR, and KIT, which are differentially expressed in SFT, although not showing any prognostic significance in primary tumours [8]. Pazopanib inhibits angiogenesis and the proliferation of tumour cells and also modifies different components of the tumour microenvironment [9]. Resistance to pazopanib has been associated with a specific immunological profile, as was observed in metastatic renal cell carcinoma, with increased levels of IL6, IL8, VEGF, and numbers of granulocytic myeloid-derived suppressor cells [10]. Moreover, an increased neutrophil/lymphocyte ratio (NLR) after pazopanib treatment has been independently associated with significantly shorter PFS and OS in some STS types different from SFT [11]. Additionally, high NLR values have been also associated with adverse survival in many solid tumours [12,13,14] and metastasis at diagnosis in STS [15]. Other peripheral inflammatory indexes, such as high platelet/lymphocyte ratio (PLR), have been associated with worse survival in STS [16] and other solid tumours [17] or high red cell distribution width (RDW), which has been associated with worse OS in solid tumours, including osteosarcomas [17,18,19,20,21].

This study aims to evaluate retrospectively the prognostic value of NLR, PLR, and RDW peripheral inflammatory indexes in a cohort of advanced SFT patients, treated within a prospective phase 2 clinical trial with pazopanib [5,6]. This study also tries to correlate changes among these peripheral inflammatory indexes and gene expression profiles in the tumour microenvironment.

## 2. Materials and Methods

### 2.1. Patients

Patients enrolled in two SFT cohorts (typical or malignant/dedifferentiated SFT), within a single-arm phase 2 trial (ClinicalTrials.gov, accessed on 15 July 2022, NCT02066285, EudraCT number 2013-005456-15; GEIS-32), were analysed retrospectively. Each cohort was composed of patients aged over 18 years who were diagnosed with unresectable or metastatic SFT (32 typical, 33 malignant, and 2 dedifferentiated SFT) in any location, and they showed progression in the previous 6 months (by RECIST and Choi criteria) and an ECOG performance status of 0–2. Each patient received an oral dose of 800 mg of pazopanib daily, only interrupted in case of disease progression (according to Choi criteria) or intolerance.

### 2.2. Peripheral Inflammatory Indexes 

Baseline values of neutrophils (10^9^/L), platelets (10^9^/L), and lymphocytes (10^9^/L) were obtained from complete blood count tests (some hours before first dose or, in some cases, within 72 h before the first dose) to determine NLR and PLR before treatment with pazopanib. RDW values at baseline were also obtained from complete blood tests and normalised by the highest value detected in its hospital. The optimal cut-off point to categorise patients between high and low NLR (3.78), PLR (242), and RDW (1.03) was calculated using the maxstat package for PFS and OS, choosing the value of OS for all indexes. PLR and NLR ratios were not available in 1 and 2 malignant SFT patients, respectively. RDW ratio was not available in 6 typical and 4 malignant SFT patients. 

### 2.3. Immuno-Oncology Assay

The gene expression of 549 human RNA transcripts involved in the innate and adaptive immune response to cancer was quantified with the immuno-oncology assay (HTG Molecular Diagnostics, Tucson, AZ, USA) in 49 formalin-fixed paraffin-embedded tumour samples (22 typical, 25 malignant, and 2 dedifferentiated SFT tumour samples). Sample preparation, assay performance, and data analysis were performed as we previously described [5,6].

### 2.4. Statistical Analysis

Time-to-event variables, PFS and OS, were estimated by Kaplan–Meier survival analysis from the treatment onset. PFS was assessed by median time and measured from treatment start date until progression or death. OS was assessed by median time and measured from treatment start date until death. 

Univariate analysis for comparing variables of interest (age, sex, tumour size at diagnosis (mm), presence of necrosis, metastasis-free interval (MFI), Eastern Cooperative Oncology Group (ECOG) performance status, number of mitoses per 10 hpf, neutrophil/lymphocyte ratio (NLR), platelet/lymphocyte ratio (PLR), and red cell distribution width (RDW)) was performed by log-rank test. Multivariate analysis was carried out according to the Cox proportional hazard regression model. Two side *p*-values lower than 0.05 were considered significant.

The regulated genes associated with each peripheral inflammatory index were determined by comparing high versus low NLR, PLR, or RDW indexes using DESeq2. Transcriptomic data were correlated with the NLR, PLR, and RDW peripheral inflammatory indexes using a generalised linear model (GLM) in R (*t*-test). 

## 3. Results

### 3.1. Patients

Peripheral inflammatory indexes were analysed in 67 SFT adult patients enrolled in the two cohorts of the GEIS-32 trial. The first cohort included 32 patients diagnosed as typical SFT, and the second cohort included 35 patients diagnosed as malignant/dedifferentiated SFT, of which only 2 were dedifferentiated. The median age was 63 years (range: 24–87), with 57% being female. Among them, 82% had metastatic disease before pazopanib initiation (Table 1). 

At a median follow-up of 16.8 months, the overall response rate (ORR) according to Choi criteria by central review was 53.7%. The median PFS was 7.4 months (95% CI 3.7–11.1), and the median OS was 49.8 months (95% CI 14.1–85.4).

### 3.2. High NLR, PLR, and RDW Are Associated with Worse Survival

Patients with pretreatment NLR values higher than 3.78 showed worse median PFS (4.5 (95% CI 1.9–7.0) vs. 10.8 months (95% CI 8.7–12.9), *p* = 0.010) and worse median OS (11.7 months (95% CI 3.5–19.8) vs. NR, *p* < 0.001) (Figure 1a and Table 2) compared with patients with lower NLR. Patients with pretreatment PLR values higher than 242 showed worse median PFS (4.5 (95% CI 2.0–7.0) vs. 10.1 months (95% CI 6.3–13.9), *p* = 0.005) and worse median OS (10.7 (95% CI 5.2–16.2) vs. 49.8 months (95% CI 14.6–85.0), *p* < 0.001) (Figure 1b and Table 2) compared with patients with lower PLR. Patients with pretreatment RDW values higher than 1.03 showed worse median PFS (4.0 (95% CI 0.9–7.0) vs. 9.8 months (7.4–12.3), *p* = 0.001), worse median OS (10.7 (95% CI 3.8–17.5) vs. 49.8 months (95% CI 9.4–90.2), *p* < 0.001) (Figure 1c and Table 2), and worse Choi response (3 (23%) vs. 25 patients (59%), *p* < 0.029) compared with patients with lower RDW.

Moreover, other clinicopathological factors showed prognostic value for either PFS or OS. In this case, the presence of necrosis (*p* = 0.002), metastasis-free interval (MFI) ≤ 8 months (*p* = 0.01), and number of mitoses higher than 3 per 10 hpf (*p* < 0.001) were significantly associated with worse PFS, while tumour size at diagnosis higher than 85 mm (*p* < 0.001) and ECOG status higher than 1 (*p* = 0.001) were significantly associated with worse OS. Neither age nor sex was a prognostic factor for PFS or OS (Table 2). Moreover, MFI ≤ 8 months (*p* < 0.001) was significantly associated with a worse Choi response.

In the multivariate analysis of inflammatory indexes and clinicopathological factors (Table 3), NLR higher than 3.78 was an independent factor for worse PFS (*p* = 0.008) and Choi response (*p* = 0.009), while RDW higher than 1.03 was an independent factor for worse PFS (*p* = 0.012) and OS (*p* = 0.001). Moreover, the number of mitoses higher than 3 per hpf was also independently associated with worse PFS (*p* < 0.001) and Choi response (*p* = 0.028), while MFI ≤ 8 months (*p* = 0.042) and ECOG status higher than 1 (*p* = 0.005) were independently associated with worse PFS (*p* = 0.042) or worse OS (*p* = 0.005), respectively.

### 3.3. Gene Expression Profiling According to High NLR and RDW

Gene expression profile was analysed in 51 tumour samples and correlated with inflammatory indexes. Tumour samples from patients with NLR values higher than 3.78 showed 12 genes differentially expressed (*TYK2*, *RIPK2*, *TRAF2*, *IKBKB*, *DUSP6*, *PTGS2*, *CCND3*, *RUNX1*, *HSPA1A*, *ELK1*, *TFRC*, *and NFATC4*, *p*-value < 0.05 and FDR < 0.05) versus patients with low NLR (Table 4). All of them were upregulated in patients with high NLR, with *PTGS2* and *RUNX1* being the genes with higher levels of overexpression (Log2FC = 1.31 and 1.47, respectively). *PTGS2* was the only gene differentially expressed (Log2FC = 1.88, *p*-value < 0.001, FDR = 0.011) in tumour samples from patients with high RDW (>1.03) versus low RDW. No significantly regulated genes were identified for high PLR versus low PLR.

## 4. Discussion

In this study, high NLR, PLR, and RDW peripheral inflammatory indexes showed a significant association with worse PFS and OS in patients diagnosed with advanced SFT and treated with pazopanib. Among the peripheral inflammatory indexes, high RDW was identified as an independent significant prognostic biomarker of worse outcome for PFS and OS, while high NLR was an independent prognostic factor just for worse OS. Importantly, these peripheral inflammatory indexes were evaluated in a series of advanced SFT patients enrolled in a prospective clinical trial that explored pazopanib as systemic treatment. Therefore, patients were strictly controlled in follow-up assessments. This phase II trial, GEIS-32 [5,6], was the first ever trial conducted in SFT and was activated at 16 European hospitals. The study included two different cohorts (formerly named typical and malignant SFT) and reported an outcome in PFS, OS, and ORR according to Choi criteria, clearly better than historical control achievement with chemotherapy. Similar to our results, high pretreatment and/or increased NLR values have been previously reported as predictive factors for a shorter PFS and OS in other types of STS treated with pazopanib, such as undifferentiated pleomorphic sarcoma and leiomyosarcoma [11,12]. Likewise, the prognostic role of NLR before treatment has been confirmed in different solid tumours [13], including STS [15]. On the other hand, high RDW has also been previously reported as a predictive factor in different tumours [21], including some types of sarcomas, such as osteosarcomas [20], but SFT patients were not included in these studies. In our study, other clinicopathological factors retrospectively explored, such as the presence of necrosis, MFI ≤ 8 months, and number of mitoses > 3/10 hpf or tumour size at diagnosis > 85 mm and ECOG > 1, were associated with statistically significant worse PFS or OS, respectively, in the univariate analysis. Moreover, MFI ≤ 8 months was associated with significantly worse Choi response. According to our results, NLR and RDW indexes may be used as prognostic biomarkers in advanced SFT patients treated with pazopanib.

Systemic inflammation, detectable through increased levels of C-reactive protein, cytokines, leukocytes, and their subtypes and hypoalbuminemia, was detected in our study with increased NLR and RDW values, which can be induced by the inflammatory response of the tumour microenvironment [22]. High NLR is a consequence of neutrophilia, which inhibits cytolytic activity of immune cells, such as lymphocytes, activated T cells, and natural killer cells. This fact might entail the reduction of T cell lymphocyte proportion in the tumour microenvironment. Additionally, high NLR has been associated with elevated infiltration of tumour-associated macrophages in the tumour microenvironment and elevated circulating cytokines (IL-1ra, IL-6, IL-7, IL-8, IL-9, IL-12, IFN-γ, IP10, MCP-1, MIP-1β, and PDGF-BB). Sometimes that may indicate M2 polarisation of tumour-associated macrophages [13]. On the other hand, high RDW reflects a high heterogeneity in the size of circulating erythrocytes, which indicates impaired erythropoiesis caused by inflammation among other stimuli (renal insufficiency or malnutrition). High RDW can be related to plasma inflammatory biomarkers, such as C-reactive protein (CRP), erythrocyte sedimentation rate (ESR), and IL-6 levels [21]. Various cytokines affect erythropoiesis (IL-6, IFN-γ, IL-1β, and TNF-α) via erythropoietin (EPO) production, inhibition of erythroid progenitors (IL-1α and IL-1β), and reduction in iron release (IL6). Together, this indicates a deleterious prognostic impact of inflamed circulating factors that might also reflect an inflamed tumour microenvironment in SFT.

The latent inflammation in the tumour microenvironment may promote tumour proliferation, survival of malignant cells, angiogenesis, and metastasis; subverts adaptive immune response, and hinders the action of chemotherapeutic agents [23,24,25,26]. In addition to tumour cells, neutrophils, macrophages, and lymphocytes infiltrated into the tumour microenvironment can contribute to amplifying inflammatory signalling through cytokines and chemokines. Neutrophils and macrophages secrete tumour-growth-promoting factors, including VEGF, HGF, IL-6, IL-8, MMPs, and elastases [12,27]. Our analysis of gene expression profiles in the tumour microenvironment identified genes implicated in inflammation or immune response, such as *PTGS2* and *TYK2*. High PTGS2 (COX2) expression, a key enzyme that catalyses the conversion of arachidonic acid to prostaglandin, was detected in SFT in a significantly positive correlation with patients expressing high NLR and RDW levels. COX2 is the inducible isoform of COX, activated by growth factors, inflammatory stimuli, or carcinogenic factors. Although it is usually undetectable in most normal tissues, it has been previously described as a poor prognosis factor in breast [28], lung [29], pancreas [30], colorectal [31], and ovarian [32] cancer, as well as in osteosarcoma [33,34,35,36]. COX2 is released by cancer-associated fibroblasts (CAFs), type 2 macrophages (M2), and cancer cells to the tumour microenvironment (TME), where it induces cancer stem cell (CSC)-like activity, proliferation, angiogenesis, inflammation, invasion, and metastasis [37]. COX2 inhibitors sensitise cancer cells to radio-/chemotherapy and could reduce the risk of metastasis [38], so it could be a target for advanced SFT. TYK2 is a nonreceptor tyrosine kinase that mediates cytokine signalling and is significantly correlated with high NLR index. TYK2 is part of the Janus kinase (JAK) family and heterodimerises with JAK1 and 2. Once activated, JAKs recruit and phosphorylate signal transducers and activators of transcription (STAT). TYK2 specifically transduces the activation of STAT1, 2, and 5 and transmits signalling of type I and II interferons (IFNs) [39]. Other STATs can also transduce type I IFN signalling in some conditions and cell types; somewhat that should be tested in SFT with STAT6, where it is constitutively activated by the NAB2–STAT6 fusion transcript. Notably, type I IFNs stimulate the transcription of over 1000 genes involved in inflammation and immune functions [40]. Not surprisingly, high peripheral inflammatory indexes indicating high levels of circulatory cytokines are detected in SFT, and these indexes showed a significant correlation with TYK2 expression. Besides, in cancer cells, TYK2 activation can lead to increase cell survival, cell growth, invasion, and resistance to chemotherapy. Both TYK2 and STATs are considered promising targets in cancer [41,42] and should be tested in SFT. All the remaining genes in Table 4 have shown some implication in inflammation or immune response whose thorough discussion is beyond the scope of this paper. However, runt-related transcription factor 1, RUNX1, which expressed the highest logarithmic fold change in the correlation with high NLR signature, will be also commented on. Bioinformatic data revealed that RUNX1 is overexpressed in cancer, and its overexpression was also linked to a worse prognosis. Importantly, RUNX1 expression was positively correlated with cancer-associated fibroblasts (CAFs) in more than 30 different cancers [43]. There is well-known evidence that several cytokines or chemokines, mainly orchestrated by transforming growth factor beta (TGFb), are involved in the conversion of fibroblasts or myofibroblasts into CAFs [44]. Some of these form a feedback loop between cancer cells and CAFs, with the latter being a relevant tumour-promoting component. Thus, RUNX1 inhibition might be also a reasonable target in SFT.

Interpretation might be affected by the low number of the patients enrolled in this prospective clinical trial, despite being a considerable number for a rare entity. NLR may be altered by different pathologies, such as hypertension, diabetes mellitus, hyperlipidemia, coronary artery disease, chronic kidney disease, heart failure, thyroid dysfunction, cerebrovascular disease, and peripheral arterial disease. RDW may also be altered by inadequate production of erythropoietin, observed in cases of undernutrition and impaired renal function. It is very improbable that these pathologies affected our results, taking into account the inclusion criteria of the trial.

## 5. Conclusions

In summary, high NLR and RDW are prognostic biomarkers of worse outcome in advanced SFT patients treated with pazopanib. The underlying tumour and microenvironment context that correlates with these peripheral inflammatory biomarkers seems to be governed, in some way, by tumour-promoting inflammation and resistance for immune response and for chemotherapy actions. The inhibition of COX2, TYK2, and RUNX1 will be tested soon by our lab team in SFT.

## Figures and Tables

**Figure 1 cancers-14-04186-f001:**
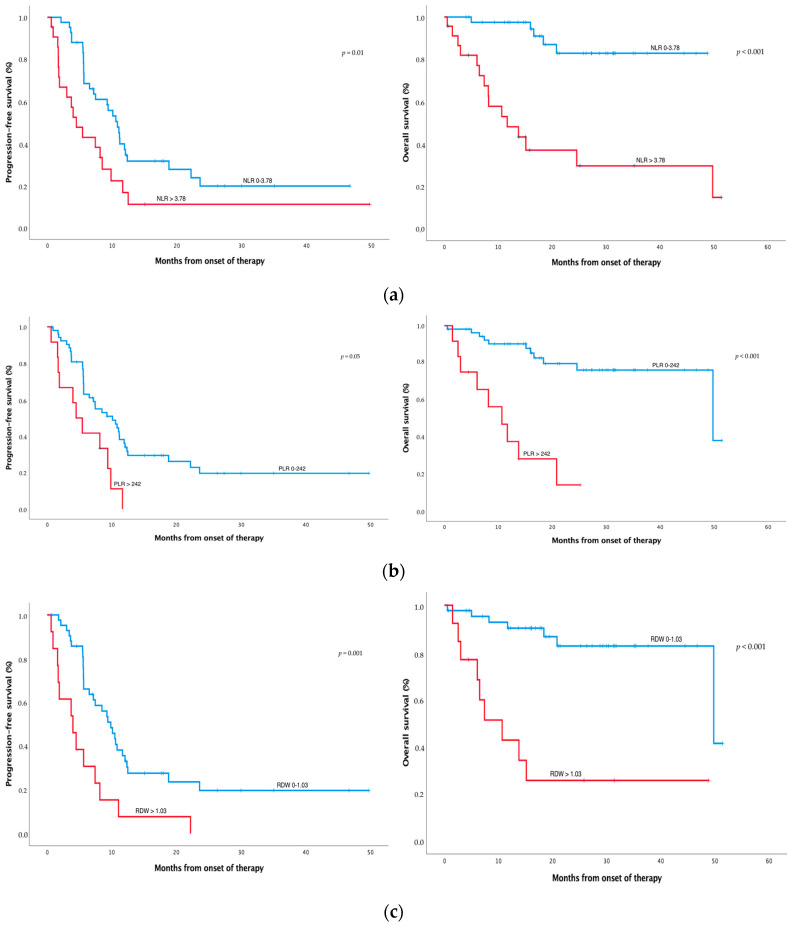
Survival analysis of SFT treated with pazopanib. Progression-free survival (PFS) and overall survival (OS), according to Choi criteria, of SFT patients with high NLR (>3.78) vs. low NLR (**a**), high PLR (>242) vs. low PLR (**b**), and high RDW (>1.03) vs. low (**c**). NLR, PLR, and RDW were measured before pazopanib treatment. The optimal cut-off point to categorise patients between high and low levels of each inflammatory index was calculated by the maxstat package. Significance between groups was defined at *p*-values < 0.05.

**Table 1 cancers-14-04186-t001:** Clinicopathologic characteristics of accrued patients.

Characteristics	Number of Cases (%)
Median age (range)	63 (24–87)
Sex (M/F)	29 (43%)/38 (57%)
ECOG (0/1/2)	52%/40%/8%
Location (primary tumour):	
Thoracic	17 (26%)
Extremities	10 (15%)
Meninges	10 (15%)
Abdominal	19 (28%)
Other	11 (16%)
Median months to M1 (range)	18 (0–302)
Histology:	
Malignant	33 (49%)
Dedifferentiated	2 (3%)
Typical	32 (48%)
Median mitotic count (per 10 hpf)	3 (0–32)
Number of mitosis (per 10 hpf):	
0–3	37 (55%)
>3	26 (39%)
Not available	4 (6%)
Metastatic/locally advanced	82%/18%

ECOG: Eastern Cooperative Oncology Group performance status.

**Table 2 cancers-14-04186-t002:** Univariate analysis of clinicopathological factors and inflammatory indexes according to progression-free survival (PFS), overall survival (OS), and Choi response.

Characteristics	Median PFS (95% CI)	*p*-Value	Median OS (95% CI)	*p*-Value	Choi Response (%)	*p*-Value
Age (years):		0.94		0.18		0.46
0–63	0.8 (6.7–12.9)		NR		20 (61%)	
>63	7.1 (3.1–11.1)		49.8 (13.6–85.9)		16 (50%)	
Sex:		0.4		0.34		1
Male	10.5 (8.5–12.5)		NR		16 (57%)	
Female	7.1 (3.9–10.4)		49.8 (NA)		20 (54%)	
Tumour size (mm) at diagnosis:						
0–85	11 (5.7–16.3)	0.14	NR	<0.001	16 (48%)	0.32
>85	8.1 (4.5–11.8)		20.8 (10.6–31)		20 (62%)	
Necrosis:		0.002		0.86		
No	10.5 (8–13)		NR		30 (58%)	0.33
Yes	4 (0.8–7.1)		49.8 (NA)		4 (40%)	
MFI (months):		0.011		0.15		<0.001
0–8	5.6 (3.5–7.8)		NR		10 (30%)	
>8	11.2 (9.2–13.2)		49.8 (14.1–85.4)		26 (81%)	
ECOG:		0.087		0.001		
0	10.6 (8.8–12.3)		NR		23 (66%)	0.085
1–2	7.4 (4.7–10.1)		18.4 (9.8–26.9)		13 (43%)	
Number of mitoses (per 10 hpf):						
0–3	11.2 (8.7–13.6)	<0.001	50 (8.7–90.8)	0.94	21 (58%)	0.61
>3	5.6 (4.4–6.7)		NR		13 (50%)	
NLR:		0.01		<0.001		0.18
0–3.78	10.8 (8.7–12.9)		NR		26 (62%)	
>3.78	4.5 (1.9–7)		11.7 (3.5–19.8)		9 (43%)	
PLR:		0.005		<0.001		
0–242	10.1 (6.32–13.9)		49.8 (14.6–85)		30 (58%)	0.35
>242	4.5 (2–7)		10.7 (5.2–16.2)		5 (42%)	
RDW:		0.001		<0.001		0.029
0–1.03	9.8 (7.4–12.3)		49.8 (9.4–90.2)		25 (59%)	
>1.03	4.0 (0.9–7)		10.7 (3.8–17.5)		3 (23%)	

MFI: metastasis-free interval, ECOG: Eastern Cooperative Oncology Group performance status, NLR: neutrophil/lymphocyte ratio, PLR: platelet/lymphocyte ratio, RDW: red cell distribution width, NR: not reached, NA: not available.

**Table 3 cancers-14-04186-t003:** Multivariate analysis of clinicopathological factors according to progression-free survival (PFS), overall survival (OS), and Choi response.

	Factor	HR	CI 95%	*p*-Value
PFS	MFI ≤ 8	2.1	1.0–4.1	0.042
	NLR > 3.78	2.8	1.3–5.9	0.008
	RDW > 1.03	2.8	1.3–6.3	0.012
	No. mitoses > 3/10 hpf	5.0	2.3–10.4	<0.001
OS	RDW > 1.03	7.4	2.4–23.0	0.001
	ECOG > 1	8.8	1.9–40.5	0.005
Choi response	NLR > 3.78	25.0	2.3–277.7	0.009
	No. mitoses > 3/10 hpf	15.4	1.4–176.6	0.028

HR: hazard ratio, CI: confidence interval, MFI: metastasis-free interval, NLR: neutrophil/lymphocyte ratio, RDW: red cell distribution width, ECOG: Eastern Cooperative Oncology Group performance status.

**Table 4 cancers-14-04186-t004:** Genes significantly expressed in patients with high NLR versus low NLR.

Gen ID	Log2FC	*p*-Value	FDR
TYK2	0.77	0.000	0.048
RIPK2	0.83	0.000	0.048
TRAF2	0.74	0.000	0.048
IKBKB	0.62	0.001	0.048
DUSP6	0.96	0.001	0.048
PTGS2	1.31	0.001	0.048
CCND3	0.85	0.001	0.048
RUNX1	1.47	0.001	0.048
HSPA1A	1.04	0.001	0.048
ELK1	0.70	0.001	0.048
TFRC	0.75	0.001	0.048
NFATC4	0.80	0.001	0.049

Log2FC: Log2 fold change (positive values—upregulated genes; negatives values—downregulated genes in high NLR vs. low NLR); FDR: false discovery rate.

## Data Availability

Data are available in a private area of the sponsor’s website (www.grupogeis.org, accessed on 15 July 2022) and will be available beginning 3 months and ending 5 years after publication of the initial study results. Data requests should be sent to secretaria@grupogeis.org.

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
