# Peer review of "Peripheral Inflammatory Indexes Neutrophil/Lymphocyte Ratio (NLR) and Red Cell Distribution Width (RDW) as Prognostic Biomarkers in Advanced Solitary Fibrous Tumour (SFT) Treated with Pazopanib"

_cancers, 2022, doi:10.3390/cancers14174186_

Round 1
Reviewer 1 Report
I note the presence of several renowned sarcoma oncologists among the authors. I feel honoured and it has been a pleasure reviewing this manuscript. Not surprisingly, it does not afford much criticism. I have only a few minor comments:
Gene expression analysis constitutes a significant part of the present work and its results are discussed in detail. The motivation for these analyses should perhaps be mentioned in the introduction.
The inflamed tumor microenvironment is complex. In the discussion, the authors provide a multitude of hypotheses generated from the inflammatory index and genetic analyses of this study. They also link it to and support by evidence from the scientific literature. I suggest that they summarize these hypotheses, i.e. the principal inflammatory pathways involved, within the discussion (as they did already in the conclusion). Just to give it more structure, to make it more understandable, not to repeat.
The authors report pre-treatment inflammatory indexes. It would be interesting how these changed under pazopanib treatment and how this change correlated to outcome parameters. Why did the authors not analyse inflammatory indexes at later times? I would assume these data were available in a clinical trial.
In a full sentence, reading „lymphocyte/neutrophil” instead of “lymphocyte-neutrophil ratio” or “lymphocyte to neutrophil ratio” seems awkward. It saves one word but does not sound correct. Same for platelet-lymphcyte ratio.
Page 3, Introduction: “Also, high NLR values have been also associated with adverse survival in many solid tumours [11–13], …” Erase one “also”.
Please change the layout of the first coloumn in tables 1 and 2 to left-bound or delete the dots, which are confusing.
Figure 1 is illegible. Please enlarge.
Page 8, discussion. “Additionally, high NLR have been associated with elevated infiltration of tumour associated macrophages in the tumour microenvironment and elevated circulating cytokines (IL-1ra, IL-6, IL-7, IL-8, IL-9, IL-12, IFN-γ, IP10, MCP-1, MIP-1β and PDGF-BB), sometimes that would indicated M2 polarization of tumour as-sociated macrophages [12].” – Long and awkward sentence. You may want tu split this before “sometimes”. Also, “Sometimes, that may indicate M2 polarization…” might be better.
Author Response
1.- Gene expression analysis have been mentioned at the end of the introduction: "This study also tries to correlate changes among these peripheral inflammatory indexes and gene expression profiles in the tumor microenvironment."
2.-We discuss about some genes, which we have previously identified, implicated in different inflammatory pathways that we consider more interesting in this process by their fold change and because they could be targeted, but we don't know which pathway could be the most important. The following changes were done to make it more understandable: "Together the previous information indicates a deleterious prognostic impact of inflamed circulating factors" was changed by "Together this indicates a deleterious prognostic impact...".This sentence "Our analysis of gene expression profile in tumour microenvironment showed that high PTGS2..." was changed by "Our analysis of gene expression profiles in the tumour microenvironment identified genes implicated in inflammation or immune response such as PTGS2 and TYK2".
3.-Inflammatory indexes after pazopanib treatment. We finally decided to analyze peripheral inflammatory indexes only at baseline because we considered the bias would be higher at post-treatment (best response or progression) by the very low N and a statistical low power. If it was required we could do this analysis but we need more time.
4.-"Neutrophil/lymphocyte" has been changed by “Neutrophil/lymphocyte ratio” in the headline and abstract. "Platelet/lymphocyte" has been changed by "Platelet/lymphocyte ratio" in the abstract.
5.-Page 3, Introduction: “Also, high NLR values have been also associated with adverse survival in many solid tumours [11–13] has been changed by “Furthermore, high NLR values have been also associated with adverse survival in many solid tumours [11–13].
6.-Layout of fist column in table 1 and 2 has been changed
7.-Figure 1 has been enlarged.
8.-Page 8, discussion. “Additionally, high NLR have been associated with elevated infiltration of tumour associated macrophages in the tumour microenvironment and elevated circulating cytokines (IL-1ra, IL-6, IL-7, IL-8, IL-9, IL-12, IFN-γ, IP10, MCP-1, MIP-1β and PDGF-BB), sometimes that would indicated M2 polarization of tumour associated macrophages [12].” has been changed by “ IP10, MCP-1, MIP-1β and PDGF-BB). Sometimes, that may indicate M2 polarization…”
9.-We consider a Graphical Abstract (GA) or a Video Abstract (VA) would be very simple in this paper and it couldn't add value.
10.-We add a new reference (ref 2): Martin-Broto, J.; Mondaza-Hernandez, J.L.; Moura, D.S.; Hindi, N. A Comprehensive Review on Solitary Fibrous Tumor: New Insights for New Horizons. Cancers 2021, 13, 2913, doi:10.3390/cancers13122913.
11.-Institutional emails of 4 authors have been added:
Dr. Samuel Hidalgo-Rios samuel.hidalgo.sspa@juntadeandalucia.esDr. Andrés Redondo andres.redondos@uam.es
Dr. Antonio Gutierrez antoniom.gutierrez@ssib.es Dra. Josefina Cruz jcrujur@gobiernodecanarias.org
12.-Grammatical correction from a native english speaker:
"malignant SFTs subtypes" by "malignant SFT subtypes"
"shown a longer progression" by "shown longer progression"
"calculated by the maxstat package" by "calculated using the maxstat package"
"tumor" by "tumour"
"indexes by DESeq2" by "indexes using DESeq2"
"In this case, presence of necrosis" by "In this case, the presence of necrosis"
"being PTGS2 and RUNX1 the gene with higher" by "with PTGS2 and RUNX1 being the genes "
"conducted trial" by "trial conducted"
"The systemic inflammation" by "Systemic inflammation"
"inflammatory response" by "the inflammatory response"
"high NLR have" by "high NLR has"
"polarization" by "polarisation"
"an impaired erythropoiesis" by "impaired erythropoiesis"
"in the tumour" by "into the tumour"
"to amplify an inflammatory" by "to amplifying inflammatory"
"where induces" by "where it induces"
"sensitizes" by "sensitise"
"and significantly" by "and is significantly"
"commented" by "commented on"
"being the latter" by "with the latter being"
Reviewer 2 Report
The Authors performed a retrospective analysis on advanced solitary fibrous tumors enrolled within GEIS-32 phase II study. The authors reported the relationship between baseline neutrophil/lymphocyte (NLR), platelet/lymphocyte (PLR) and red cell distribution width (RDW) and prognostic parameters such as PFS, OS and Choi response.
Given the rarity of tumors and the sample size available, the results described may be of interest for readers involved in management of this kind of tumor.
Author Response
Thank you very much for your opinion about our work.